# Characterization of Polycaprolactone Nanohydroxyapatite Composites with Tunable Degradability Suitable for Indirect Printing

**DOI:** 10.3390/polym13020295

**Published:** 2021-01-18

**Authors:** Stephanie E. Doyle, Lauren Henry, Ellen McGennisken, Carmine Onofrillo, Claudia Di Bella, Serena Duchi, Cathal D. O’Connell, Elena Pirogova

**Affiliations:** 1Electrical and Biomedical Engineering, School of Engineering, RMIT University, Melbourne, VIC 3000, Australia; s3540745@student.rmit.edu.au (S.E.D.); lhenry.lauren@gmail.com (L.H.); ellen.mcgennisken@gmail.com (E.M.);; 2BioFab3D@ACMD, St Vincent’s Hospital Melbourne, Fitzroy, VIC 3065, Australia; carmine.onofrillo@unimelb.edu.au (C.O.); claudia.dibella@unimelb.edu.au (C.D.B.); serena.duchi@unimelb.edu.au (S.D.); 3Department of Surgery, The University of Melbourne, St Vincent’s Hospital Melbourne, Fitzroy, VIC 3065, Australia; 4ARC Centre of Excellence for Electromaterials Science, Intelligent Polymer Research Institute, University of Wollongong, Wollongong, NSW 2522, Australia; 5Department of Orthopaedics, St Vincent’s Hospital Melbourne, Fitzroy, VIC 3065, Australia

**Keywords:** composite materials, bone regeneration, degradable scaffold, PCL, HA

## Abstract

Degradable bone implants are designed to foster the complete regeneration of natural tissue after large-scale loss trauma. Polycaprolactone (PCL) and hydroxyapatite (HA) composites are promising scaffold materials with superior mechanical and osteoinductive properties compared to the single materials. However, producing three-dimensional (3D) structures with high HA content as well as tuneable degradability remains a challenge. To address this issue and create homogeneously distributed PCL-nanoHA (nHA) scaffolds with tuneable degradation rates through both PCL molecular weight and nHA concentration, we conducted a detailed characterisation and comparison of a range of PCL-nHA composites across three molecular weight PCLs (14, 45, and 80 kDa) and with nHA content up to 30% *w*/*w*. In general, the addition of nHA results in an increase of viscosity for the PCL-nHA composites but has little effect on their compressive modulus. Importantly, we observe that the addition of nHA increases the rate of degradation compared to PCL alone. We show that the 45 and 80 kDa PCL-nHA groups can be fabricated via indirect 3D printing and have homogenously distributed nHA even after fabrication. Finally, the cytocompatibility of the composite materials is evaluated for the 45 and 80 kDa groups, with the results showing no significant change in cell number compared to the control. In conclusion, our analyses unveil several features that are crucial for processing the composite material into a tissue engineered implant.

## 1. Introduction 

Large-scale bone loss can occur as a result of trauma, tumour removal, or infection and results in a defect that cannot spontaneously heal [1]. With over two million surgeries performed worldwide a year, bone grafts remain the gold standard for the treatment of bone loss [2]. 

While metal prosthetic implants are often successful, their use is limited when the bone stock (surrounding bone to anchor the implant) is inadequate. Additionally, there are still significant challenges associated with implant incorporation, failure, component wear, and infection [3,4]. These limitations are motivating research toward a new tissue engineering strategy, whereby a degradable implanted scaffold would trigger the body to repair natural bone within the area of the implanted scaffold [5]. With this approach, the ability to tune the rate of degradation is critical to ensure there is sufficient concomitant bone formation to withstand the mechanical loads within the body before the scaffold has reached a critical degradation point. If the scaffold degrades too fast, it may cause a failure of the implant. Instead, if the degradation rate is too slow, the scaffold does not allow a sufficient space for the new bone to mature or to integrate within the existing tissue [6]. Bone has the capacity to regenerate and fill small bone defects commonly through the process of secondary healing by endogenous progenitor cells [4,7]. However, critical size defects (>1–2.5 cm) cannot heal without therapeutic aids or materials designed to encourage bone regeneration [8]. Several approaches, such as the bone graft substitutes, have focussed on improving the efficacy of bone grafts or other scaffolds by incorporating bone progenitor cells and growth factors to stimulate cells [4]. An ideal bone graft or scaffold should be made of biomaterials that imitate the structure and properties of a natural bone extracellular matrix, including osteoprogenitor cells, and provide all the necessary environmental cues found in a natural bone [9]. Various tissue engineering solutions involve the incorporation of a natural or a synthetic bone additive to enhance the osteoinductive (promoting undifferentiated cells down a bone-forming lineage) and osteoconductive (promoting bone growth on an implant) properties of a material to achieve increased osteogenesis [10]. 

Among the osteogenic compounds, hydroxyapatite (HA) is a calcium phosphate mineral naturally present in the bone in the form of nanoscale crystals [11]. Nanoparticle HA (nHA) can also be synthesised in the form of a powder, but it needs to be incorporated within a carrier material to serve as a functional scaffold or implant within the body [12]. The role of the carrier is to house the bone additive and make the entire material processible into a 3D structure. Carriers can include polymers, metals, hydrogels, ceramics, etc. in order to be used directly in 3D printing (fused deposition modelling (FDM), melt electrowriting (MEW), selective laser sintering), freeze drying, sol–gel, casting, foaming, or even as a coating [13,14,15,16,17,18,19,20]. Of the wide range of carrier materials available, polycaprolactone (PCL) is of significant interest due to its biodegradability and status as an approved implant material by many regulators, including the US Food and Drug Administration (FDA). PCL is used in a wide range of tissue engineering applications, including breast implants, skin grafts, bone and dental regeneration, and vascularisation [21]. PCL is available in a wide range of molecular weights, ranging from 530 to 120,000 Da, thus providing a useful vector for tuning its properties, including degradation rate and Young’s modulus [15,22]. The addition of bone additives to the PCL, such as HA, changes its key properties affecting processability, such as the viscosity; however, the extent of these changes depends on the original material and both the type and concentration of incorporated bone additives. The ability to understand the final composite material can allow for the tailoring of a material towards a particular fabrication technique, mechanical, and/or biological application. In bone regeneration, this may be driven by the size, type, or location of the defect to be repaired. Here, we thoroughly characterise three different molecular weights of PCL (14,000, 45,000, 80,000 Da) with and without varying concentrations of synthetic nHA up to 30% *w*/*w*. Molecular weight is a controllable property influencing the rate of degradation of an implant compared with factors such as the site of implantation and applied stress, which there is often little to no control over [23]. Therefore, we studied the various combinations of the two materials to evaluate if the degradation rate of the composite material could be tuned via PCL molecular weight and/or nHA concentration. Further analysis reveals the rheological properties, homogeneity of the nHA dispersion, mechanical properties, 3D fabricability, and cytocompatibility, which are all features that are crucial for processing the composite material into a tissue engineered implant with the desirable properties. 

## 2. Materials and Methods

### 2.1. Material

PCL of three different molecular weights were used: 14, 45, and 80 kDa (Sigma-Aldrich, St. Louis, MO, USA). nHA with particle size <200 nm was used (Sigma-Aldrich, St. Louis, MO, USA).

### 2.2. Preparation of the PCL-nHA Composite 

The mixing of each PCL with the various concentrations of nHA followed the protocol outlined by Abdal-Hay et al. [15]. Briefly, PCL was dissolved in chloroform (Chem-Supply, Gillman, SA, Australia) at 15% *w/v* by vigorous stirring for 40 min. nHA was added in 10 mg batches with vigorous stirring for 5 min in between the batches. After all the nHA was added, the solution was ultrasonically agitated for 20 min followed by the vigorous stirring overnight. The next day, the solution was again ultrasonically agitated for 30 min before being cast in a petri dish and placed under vacuum for 3 days to allow the chloroform to fully evaporate. 

### 2.3. Thermogravimetric Analysis (TGA)

The final concentration of nHA in the uncasted PCL-nHA composite material was verified using a TGA 8000™ Thermogravimetric analyser (Perkin Elmer, Waltham, MA, USA). Samples between 3 and 6 mg were heated at 10 °C per min from 30 to 850 °C under air at 20 mL/min. For each test group, a technical quadruplicate (n = 4) was used. 

### 2.4. Rheology

The viscosity of the uncasted 14, 45, and 80 kDa PCLs with and without nHA were measured using a Physica MCR 302 Rheometer (Anton Paar, Graz, Austria) in a cone-plate geometry (15 mm diameter with a cone angle of 1°, and a truncation of 31 µm) at 70 °C and a shear rate of 1 s^−1^. Shear ramp measurements showed that the viscosity of molten PCL or PCL-nHA at 70 °C was independent of shear rate in the relevant range of 0.1 to 100 s^−1^. For each test, group a technical quadruplicate (n = 4) was used. 

### 2.5. Mechanical Testing

The Young’s modulus was assessed via compression tests performed on cylindrical disks of material (7 mm diameter, 2 mm height) at room temperature using a TA Electroforce 5500 mechanical loading device (TA Instruments, New Castle, DE, USA) fitted with a 50 lb load cell. The diameter of each sample was taken prior to testing. Samples were compressed between two stainless steel plates in an unconfined setting. The bottom plate was in a fixed position and the top plate moved following a ramp function at a rate of 0.01 mm s^−1^ until a total displacement of 10% of the sample height or until maximum machine capabilities (200 N). Load and displacement measurements were recorded and converted into stress (σ) and strain (ε) data using the measured cross-sectional area of the disk and its height. The stiffness was then computed using the linear region of the stress–strain curve. For each test group, a technical quadruplicate (n = 4) was used. 

### 2.6. Scanning Electron Microscope (SEM) Imaging

SEM imaging was carried out on the uncasted 30% nHA composite groups. Samples were imaged using the FEI Quanta 200 SEM (Thermo Fisher Scientific, Waltham, MA, USA) under high vacuum mode, 5 spot size, and 25 kV accelerating voltage. To increase the contrast of the SEM images, a Fiji plugin, Enhance Local Contrast (CLAHE) filter, was applied using the following settings: block 50, bin 256, and slope 3 [24]. 

### 2.7. Accelerated Degradation

Due to the slow degrading nature of PCL (>2 years in vivo) [25], an accelerated test based off that by Lam et al. [26] was used to achieve comparative results between the groups. Briefly, the same solid cylindrical disks used for the compression testes (7 mm diameter, 2 mm height, 0.08–0.13 g) were submerged in 4 mL of 5 M sodium hydroxide (NaOH) and maintained at 37 °C. After 6 hours, the disks were removed, washed in phosphate buffer solution (PBS), and placed in a dehydrator (Food Lab™ Electronic Dehydrator, Sunbeam, Florida, USA) overnight at 35 °C. Then, the weight was recorded. After a total of 48 h in NaOH (8 × 6 h), the NaOH soaking time was increased to 1 × 24 h with all other steps were kept constant. For each test group, a technical quadruplicate (n = 4) was used. 

### 2.8. Scaffold Fabrication 

Logpile structures (8 mm^3^ with 480 μm strut diameters and 1200 μm spacing) were fabricated by indirect 3D printing. Briefly, moulds were 3D printed and then filled with the PCL or PCL-nHA composite materials before the moulds were removed. 

### 2.9. Alizarin Red S Staining 

Alizarin Red S (Sigma-Aldrich, St. Louis, MO, USA), used for the detection of calcified matrix deposition, was employed as a qualitative assay. First, previously prepared logpile structures were placed in a 24-well plate with 1.5 mL of 40 mM Alizarin Red S added to cover the structure. The well plate was covered in foil to limit exposure light and incubated for 40 min at room temperature under agitation. Then, the excess of Alizarin Red was removed from each well, and scaffolds were washed 10 times in distilled water. Then, 1.5 mL of PBS was added to cover each structure before again covering in foil and incubating for 30 min at room temperature under agitation. PBS was removed, structures given 10 min to dry, and then macroscopic pictures of all structures were taken. 

### 2.10. Cytocompatibility Assay

Cylindrical disks (7 mm diameter, 2 mm height) of PCL controls and PCL-nHA were cleaned via passive 7-day soaking in PBS with PBS changed daily. After the 7-day soak, disks were sterilised by soaking in ethanol for 1 h followed by UV light treatment for 1 h. To remove any ethanol residues, disks were washed for 5 × 5 min in sterile PBS. Following this, disks were soaked in 5 mL of complete proliferation media for an additional 7 days with the conditioned media collected at the end of the 7 days. This conditioned media was used to test cell viability on the commercial MG-63 osteoblastic cell line (ATCC^®^ CRL-1427™, ATCC, Manassas, VA, USA) [27]. Briefly, cells were plated at 20,000 cells per well in a 24-well plate and cultured in fresh complete cell culture media; Dulbecco’s Modified Eagle’s Medium-high glucose (DMEM-HG, glucose 4500 mg/L, Sigma Aldrich) containing 10% FBS (Gibco, Thermo Fisher Scientific, Massachusetts, MA, USA), 100 U mL^−1^ Penicillin, and 100 µg mL^−1^ Streptomycin solution (Gibco), 2 mM L-Glutamine (Gibco), and 15 mM HEPES (Gibco) for 24 h at 37 °C, 5% CO_2_ air. After 24 h, the media was replaced with the conditioned media for an additional 24 h at 37 °C in a humidified atmosphere of 5% CO_2_ air. CellTiter Blue (Promega, Wisconsin, WI, USA) metabolic assay was then applied following the manufacturer’s instructions and incubating for 2 h at 37 °C, 5% CO_2_ air. The control group was fresh complete cell culture medium. Then, 100 µL of the surnatant was transferred to a 96-well plate, and the entire well plate was read at 550-15/600-20 emission/excitation fluorescence wavelengths using a CLARIOstar plate reader (BMG Labtech, Ortenberg, Germany). Cell number was calculated based off a standard curve of cells with a good fit of linearity at R^2^ = 0.97782. For each test group, a biological quadruplicate (n = 4) was used. 

### 2.11. Statistical Analysis 

For each experiment and test group, four technical or biological replicates (n = 4) were used with data summarised as the mean with error bars representing standard deviation. Statistical analysis was performed using Prism 8 (GraphPad, California, USA) with a statistical significance level ≤ 0.05. A one-way analysis of variation (ANOVA) was used to determine statistical significance between groups. In all graphs, stars represents the following: * is *p* ≤ 0.05; ** is *p* ≤ 0.01; *** is *p* ≤ 0.001; **** is *p* ≤ 0.0001. All results are not significant unless marked with a star/s or otherwise labelled. 

## 3. Results and Discussion

### 3.1. Preparation and Verification of PCL-nHA Composite

In this work, we have used a chloroform solution method to generate PCL-nHA composites with well-dispersed nHA particles. We mixed a range of composite materials made of PCL at three different molecular weights (14, 45, and 80 kDa) and four different concentrations of nHA (0, 10, 30, and 50% *w*/*w*). Then, the final amount of nHA in the composite was quantified through Thermogravimetric Analysis (TGA). nHA is a mineral with a melting point over 1000 °C, whereas PCL has a low melting point at around 60 °C [20,21]. Therefore, PCL burns off during the 30–850 °C temperature sweep, whereas nHA is stable, allowing the accurate measurement of both the nHA and PCL concentrations. Figure 1a shows the control groups with 0% nHA as expected. With the intended 10% nHA groups, both the 14 kDa and 80 kDa PCL show a slight decrease from the expected concentration at 8.53 ± 0.73% and 9.08 ± 0.43% respectively, while the 45 kDa PCL group showed a significant increase compared to the other two 10% nHA groups at 14.02 ± 0.54%. The intended 30% nHA groups had a final concentration of 22.43 ± 0.22%, 22.74 ± 1.67% and 21.52 ± 2.17% for the 14, 45, and 80 kDa PCL groups, respectively. Finally, for the intended 50% nHA group, the final concentrations were 30.61 ± 0.71%, 33.40 ± 0.75%, and 30.85 ± 0.53% for 14, 45, and 80 kDa PCL groups, respectively. The highest concentration achieved (approximately 33%) is about half that of the maximum achievable particle concentration (64%) for randomly packed spherical particles. It was found that the greater the amount of nHA being initially added, the more of it was lost during the mixing process (Figure 1b), which is most likely due to the transfer between glassware. Apart from the abnormality seen in the 45 kDa 10% nHA group, the drop in the nHA content was consistent in each concentration group regardless of the PCL molecular weight. Amongst the studies producing the highest concentration PCL-HA 3D scaffolds, Li et al. created the PCL-nHA electrospun mats at the concentrations of up to 60%, while both Ba Linh et al. and Chuenjitkuntaworn et al. created 50% (*w/v* and *w*/*w* respectively) nHA composite scaffolds [28,29,30]. However, despite claims of the high nHA concentration, none of the studies verified the intended or added concentration and therefore, the true final concentration is unknown. Of the few papers that did perform concentration verification, Ang et al. achieved the maximum final concentration of microscale-HA (mHA) consistent with our data at 29.84%, while Trakoolwannachai et al. also experienced large mHA loss with their intended 30% mHA-PCL scaffold, only having a final concentration of 13.12% mHA [31,32]. 

From here on, the nHA concentration will be referred to by the measured concentration rather than the theoretical or intended range: 10% as 10% (±4.57%), 30% as 20% (±5.6%), and 50% as 30% (±4.3%). 

### 3.2. Viscosity Measurement of PCL-nHA Composite

Understanding the rheological properties of scaffold composites is critical for optimising the fabrication processes, especially in the nozzle-based 3D printing techniques. The molecular weight of a polymer is described by the average length of the polymer chains and with increasing chain lengths, the viscosity also increases [33]. This is seen with the pure PCLs where the viscosity increases from 38.7 ± 0.3 to 1131.5 ± 6.5 to 47,735 ± 3123 Pa.s for the 14, 45, and 80 kDa PCL, respectively. Within each PCL group, as the concentration of nHA increased, so did the viscosity (Figure 2). Our results are consistent with Xue et al., who also saw an increase in the viscosity when nHA between 0 and 7% was mixed with 80 kDa PCL [34]. 

### 3.3. Stiffness Evaluation of PCL-nHA Composite

For bone scaffolds designed for implantation within the body, the mechanical properties of the scaffold should be comparable to the bone it is replacing, and hence, they need to be evaluated to ensure, at minimum, the implant does not fail. The final mechanical properties of a scaffold are equally dependent on the design and the selected material [35]. Relevant mechanical properties for bone implants include the compressive and tensile strength, fracture toughness, sheer modulus, stiffness, fatigues strength, or failure mechanism as well as dynamic mechanical analysis. In our study, we focus on compressive stiffness only as a proxy to compare the relative mechanical properties across a range of compositions.

Therefore, we assessed the material suitability aspect by analysing how the addition of nHA affects the Young’s modulus of the bulk uncasted materials. Both the 14 kDa and 80 kDa molecular weight groups exhibited the same trends in result. First, there is a drop in the Young’s modulus from the control group (0% nHA) to the lowest nHA concentration group (10% nHA); however, it was seen that increasing the concentration of nHA also increased the Young’s modulus (Figure 3). Despite it being well established in the literature that adding HA to PCL increases the mechanical properties of a composite, Baji et al. also observed that adding HA initially decreased the stiffness of PCL alone, while then with the increasing concentrations of HA, the Young’s modulus also increased [36]. Rajzer also reported a significant decrease in the tensile strength of the PCL-nHA scaffolds compared to only PCL (control) [37]. On the other hand, the 45 kDa group displays no obvious trend. It is also noted that there is a high degree of variation within the technical quadruplicates as seen by the size of the error bars, and there are no significant differences within in any PCL group, and therefore, the trends seen can be considered weak. 

### 3.4. Distribution of nHA Particles in PCL Composites 

Achieving a homogenous distribution of the nHA in the PCL, both before and after the fabrication procedure, is mandatory for a consistent biological performance during the application. As nHA is the osteoinductive component of the material, if it is unevenly distributed through the final product, it may cause uneven cell attraction, matrix, and mineral deposition and even unwanted localised bone formation [38,39]. From a 3D printing perspective with a nozzle-based fabrication technique, non-evenly dispersed material or large agglomerates can cause clogging within the nozzle and either completely block or cause inconsistent printing [40]. Mixing the nHA particles with the addition of chloroform minimises agglomeration and aims to produce a homogenous dispersion of the nHA [14]. To verify this, SEM images of the highest concentration group of nHA in each PCL were taken to see how the nHA was distributed. Figure 4 shows even distribution of nHA (white dots) in all the PCL molecular weight groups. Small agglomerates are evident in the 45 kDa PCL and 80 kDa PCL groups (red arrows). The absence of agglomerates in the image in the 14 kDa group does not imply that there are none present in any of the samples, although it may suggest the lower molecular weight of PCL allows for easier dispersion of nHA in the solution. Nevertheless, even the two “large” agglomerates in the 45 kDa PCL group measure ≈20 µm diameter and therefore may not affect the ability to 3D print, depending on the printing technique or nozzle size used. 

### 3.5. Degradability of nHA-PCL Composites 

PCL can degrade via hydrolysis or enzymatic breakdown. Both mechanisms produce similar degradation products which can be removed from the body through natural processes [41]. Theoretically, the greater the molecular weight, the slower the degradation, as there are longer chains with more ester bonds to be cleaved until the generation of water soluble monomers/oligomers that can be expelled [42]. An accelerated test based off that by Lam et al. [26] was used to achieve comparative results between the groups. Lam et al. showed that porous 80 kDa PCL logpile structures degraded approximately 80% of their weight in 4 weeks through accelerated degradation yet when PCL was combined with 20% wt tricalcium phosphate, while the accelerated degradation of the same size scaffold took only 48 h. Moncal et al. also conducted an accelerated degradation study for 72 h—however, using 0.1 M NaOH on their 70 kDa PCL or PCL/PLGA/mHA logpile scaffolds [43]. After 72 h, the PCL only displayed little mass loss (<1%), whereas the composite scaffolds had an approximately 37% drop in mass. These results show when incorporated with additives, the degradation rate of PCL drastically increases from weeks to hours or days when using an accelerated protocol. 

Across all materials, our results are consistent with the literature, which suggest the PCL degradation occurs in two steps: the first is a slow decrease in mass due to the degradation of the amorphous regions, followed by a rapid weight loss as the crystalline regions are fractured (Figure 5) [44]. While there is no direct translation from the accelerated protocol to the in vivo scenario, from our results, it can be concluded that the presence of nHA speeds up the degradation process with the highest nHA content material degrading the fastest in each case (Figure 5a–c). The increased rate of degradation, when nHA is introduced, may be due to a change in the surface area occurring due to the polymer–solvent access. With the increased nHA content, the permeability of the NaOH through the polymer also increases, so it is not acting only at the surface [26,42]. Again, as seen with the compression results, the 45 kDa PCL group responded differently—a slower rate of degradation is observed compared with the 80 kDa group, even with smaller polymer chain lengths. The representative macroscopic images (Figure 5d) generally show that the cylindrical disks retain their circularity during degradation, thereby indicating uniform degradation. This is except for the fastest degrading samples (14 kDa 20%, 14 kDa 30%, and 80 kDa 30%), which are no longer perfectly circular at their final time points. This may indicate a degree of disintegration in the final stages of degradation or may be due to manual handling and the associated damage of the crumbly samples at their final time points. 

Despite this, in order to tune the degradation rate of an implant, the molecular weight of the PCL is to be chosen as a baseline with the understanding that the addition of the nanoparticles is likely to increase the rate of degradation. As the degradation is based on the bulk property of the material and factors, such as the porosity of a scaffold, would further affect the degradation rate both in vitro and in vivo. 

### 3.6. Scaffold Fabrication and Homogeneity Analysis Post Indirect 3D Printing

The ability to convert a material or a composite into a 3D structure allows it to act as a structural and functional implant within the body. Nozzle-based 3D printing techniques, such as FDM, require laborious optimisation for the printing of new materials, such as PCL-nHA, including the fine-tuning of the rheological properties, printing speed, temperature, etc. On the other hand, indirect 3D printing involves the generation of a 3D-printed sacrificial mould which is then filled with another material. Indirect printing is advantageous over nozzle-based extrusion techniques, as it requires a fraction to none of the same optimisation as the material only needs to fill out the mould in the same way cake batter is poured into a tin. Logpile structures were created via indirect printing using all PCL-nHA groups. The 14 kDa PCL was inconsistent during fabrication with some samples crumbling (Figure 6). This was unrelated to the presence or different concentrations of nHA, with even the 14 kDa PCL only group completely crumbling in certain tests. The brittle nature of the 14 kDa material means that it is unlikely to be suitable in any application as a bone regeneration scaffold. Both the 45 kDa and 80 kDa PCL groups were able to be fabricated and did not face the same challenges as the 14 kDa group (Figure 6). The quality and consistency of the fabricated scaffolds within the 45 kDa and 80 kDa groups, regardless of the nHA concentration (up to 30% *w*/*w*), are superior to most reported in the literature. Electrospun PCL-nHA scaffolds are consistently fabricated at <10% *w*/*w* nHA and are limited to a few millimetres in the maximum height of the scaffold [34,45,46]. FDM or plotting can routinely achieve higher concentrations at 20% *w*/*w* nHA and can also achieve taller structures compared to electrospun structures but with lower porosities [47,48].

To verify that the nHA remained homogeneously distributed throughout the PCL after the indirect 3D printing process, Alizarin Red S staining was applied to all logpiles. Alizarin Red S is used as a biochemical assay to detect the presence of calcific deposition by cells of an osteogenic lineage. As nHA is a calcium phosphate, the Alizarin Red S stains the nHA particles within the PCL-nHA structures [49]. As seen in Figure 7, the control logpiles turn a light-yellow colour in the absence of nHA, while all PCL-nHA composites are stained red. Of note, only the presence of nHA affects this process and not the molecular weight of the PCL. The even colouring of the Alizarin Red S throughout the logpile structure further verifies that even after the materials have been fabricated, the nHA remains homogeneously distributed throughout.

### 3.7. Cytocompatibility Analysis of PCL nHA Composites

PCL and nHA are both well-established FDA-approved biomaterials and are the only two materials that should be present in the final form of the material. Considering the presence of chemical compounds used during the fabrication procedure, we verified whether the release of residuals from the fabricated scaffolds may affect cell viability in vitro. The MG-63 human osteoblastic cell line has been shown to be stable in cell morphology, expression of adhesion receptors, cell cycle phases, and intracellular signalling over 30 passages and as such, it can be used in the early stages of testing the interaction between cells and materials any cytotoxic effects [27]. Thus, a metabolic assay was performed with MG-63 cells to assess the viability of the cells when cultured in the conditioned media from the PCL-nHA structures. This is in line with the International Organization for Standardization 109993-5 on the Biological evaluation of medical devices tests for in vitro cytotoxicity standards, which recommend three categories of tests for assessing cytocompatibility rather than one single test, i.e., extract, direct contact, and indirect contact [50]. The extract class of the test is used for detecting toxicity in soluble substances [51]. This will allow any toxic substances that have leached into the media during the 7-day soak to be detected. The results (Figure 8) show no significant change in viability in any of the PCL-nHA groups when compared to the control group, apart from the 14 kDa 30% nHA group. The results from the accelerated degradation data show the 14 kDa 30% nHA group is the group that degrades the fastest, and this was also confirmed in the cytocompatibility preparation. By the end of the second week for the disks being in solution, the media from the 14 kDa 30% nHA group changed from the standard red colour to a cloudy orange, therefore suggesting the disks had started to degrade during this time. The media from all other groups remained clear and red. These results suggest the build-up of the degraded material was toxic to the cells, but it cannot be concluded if it was only a build-up of the degraded PCL and nHA or whether there were trace amounts of chloroform left [52]. Furthermore, Bosworth and Downes stated that the “local generation of carboxylic acid by-products, if not removed, can increase the surrounding acidity promoting autocatalysis, which subsequently accelerates the rate of degradation”, which could also suggest that the degrading material created an acidic environment that could contribute to the cell death displayed [41].

## 4. Conclusions

In this work, we aimed to characterise and compare three different PCLs to cover a wide molecular weight range (14–80 kDa) along with the incorporation of increasing concentrations of nHA, up to 30% *w*/*w*. We believe that such an investigation has not been previously completed. This work will be a reference to guide further bone tissue engineering studies for better understanding and hence selection of a particular PCL molecular weight with and without additives, when designing an improved scaffold material. The use of different molecular weights of PCL was utilised in order to create a material with the tuneable degradation rate. With increasing concentrations of nHA, the rate of degradation also increased in all PCL molecular weight groups. The combination of poor fabricability and potentially problematic cell cytocompatibility results renders the 14 kDa PCL with or without nHA unsuitable for implantation. On the other hand, the 45 and 80 kDa composite material groups were able to be fabricated by indirect 3D printing and showed that the nHA was evenly distributed after being fabricated. These two PCL groups also showed no significant change in cell number compared to the control group as evident by the cytocompatibility test deeming them suitable materials for tissue engineered bone scaffolds. However, further studies are required to understand the interaction between the nanoparticle and polymer, especially in the 45 kDa group where there was a slower rate of degradation compared to the 80 kDa group despite shorter chain lengths. Additionally, the relationship between nHA concentration and an enhanced production of mineralised matrix by osteoblastic cells is yet to be studied, which will dictate if the material is suitable for bone scaffolds.

## Figures and Tables

**Figure 1 polymers-13-00295-f001:**
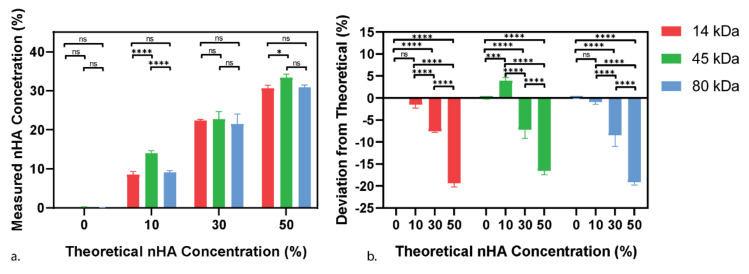
Thermogravimetric analysis of polycaprolactone nanoparticle hydroxyapatite (PCL-nHA) composites. (**a**), Final concentration of nHA in the various molecular weight PCLs; (**b**), deviation from the expected or amount of nHA initial added to the composite (% *w*/*w*). In all cases, excluding the 45 kDa 10% nHA group, there is a proportional nHA loss during the mixing process with the more nHA added, the more that is lost. Statistical significance as followed: * is *p* ≤ 0.05; *** is p ≤ 0.001; **** is *p* ≤ 0.0001.

**Figure 2 polymers-13-00295-f002:**
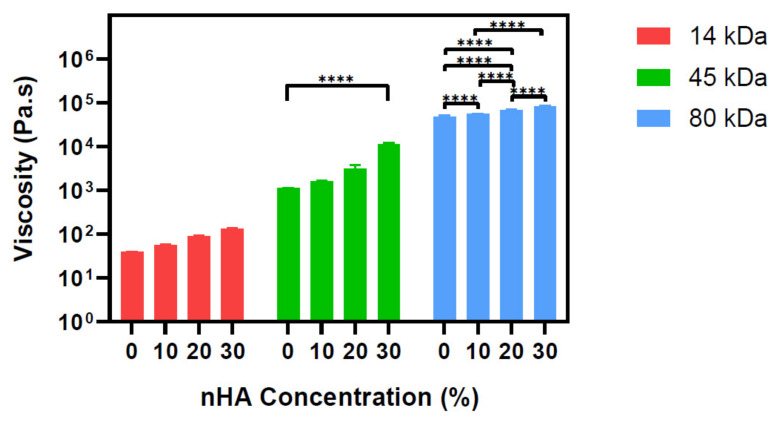
Viscosity analysis of PCL-nHA composites performed with rheology test. The molecular weight affects the viscosity of the pure PCL; however, regardless of the molecular weight, the addition of nHA increases the viscosity of the composite. Viscosity was measured at 70 °C. Statistical significance as followed: **** is *p* ≤ 0.0001.

**Figure 3 polymers-13-00295-f003:**
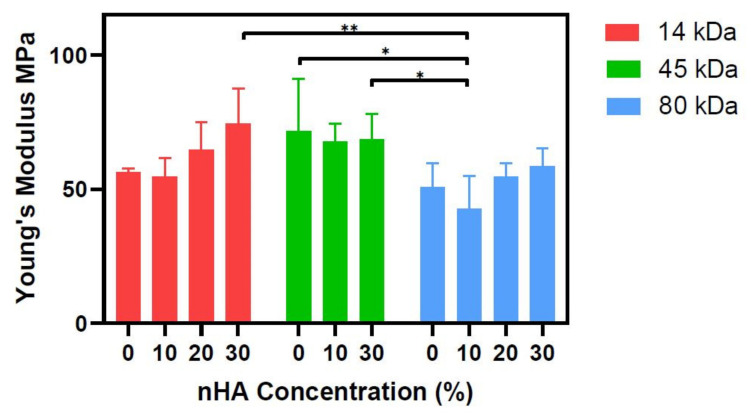
Young’s modulus of the PCL-nHA composites. Young’s modulus was calculated from the linear region of the stress/strain curved obtained from an unconfined compression test on cylindrical disks of the PCL/PCL-nHA composites. Statistical significance as followed: * is *p* ≤ 0.05; ** is *p* ≤ 0.01.

**Figure 4 polymers-13-00295-f004:**
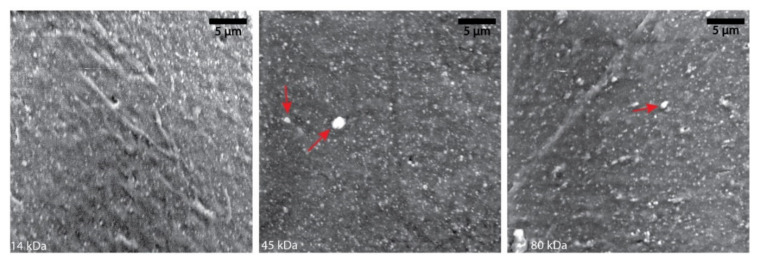
Homogeneity of the PCL-nHA composites. Representative pictures of SEM analysis showing that an even distribution of nHA is seen across all three molecular weight PCL groups. Agglomerates of various sizes are evident in the 45 and 80 kDa groups (red arrows).

**Figure 5 polymers-13-00295-f005:**
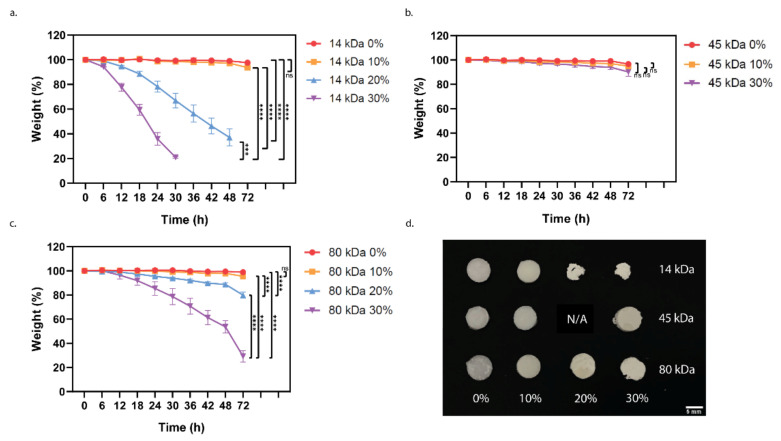
Degradation of the PCL-nHA composites measured via accelerated degradation protocol. (**a**), 14 kDa PCL with 0–30% *w*/*w* nHA; (**b**), 45 kDa PCL with 0–30% *w*/*w* nHA; (**c**), 80 kDa PCL with 0–30% *w*/*w* nHA. The cylindrical disks composites were weighed at every time point or until they no longer held one single piece. (**d**), representative macroscopic images of the cylindrical disks at their final time point (72 h for all excluding 14 kDa 30% nHA (30 h) and 14 kDa 20% nHA (48 h). Statistical significance as followed: *** is *p* ≤ 0.001; **** is *p* ≤ 0.0001.

**Figure 6 polymers-13-00295-f006:**
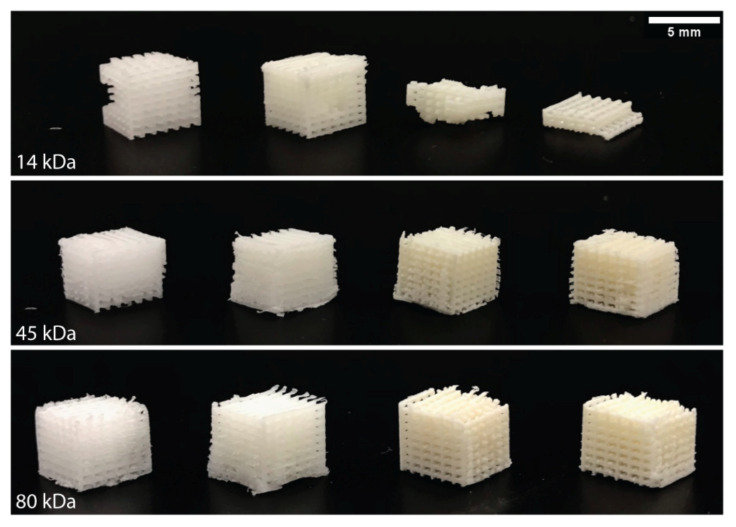
Scaffolds fabrication of PCL and PCL-nHA. Representative pictures of the indirect 3D printed logpiles using different nHA percentages. For all PCL groups, the nHA concentration increases from left to right: 0% nHA, 10% nHA, 20% nHA, and 30% nHA.

**Figure 7 polymers-13-00295-f007:**
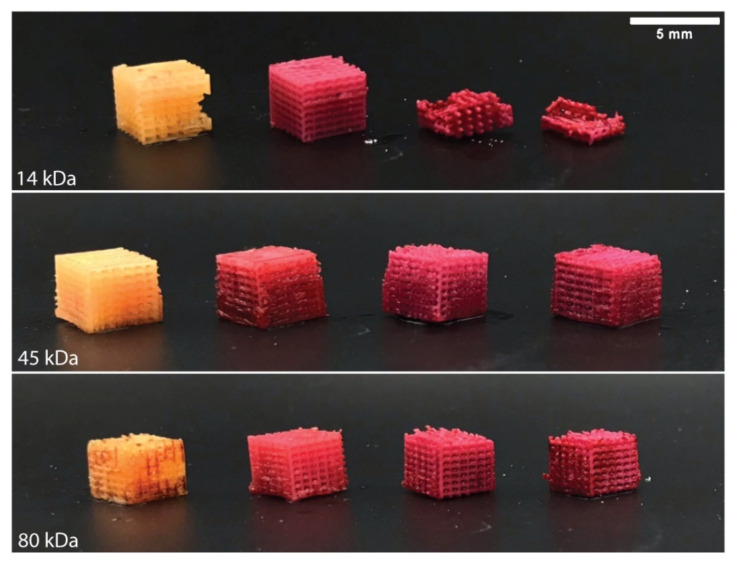
Homogeneity analysis post indirect 3D printing. Representative picture of indirect 3D printed PCL and PCL-nHA logpiles stained with Alizarin Red S. For all PCL groups, the nHA concentration increases from left to right: 0% nHA, 10% nHA, 20% nHA, 30% nHA.

**Figure 8 polymers-13-00295-f008:**
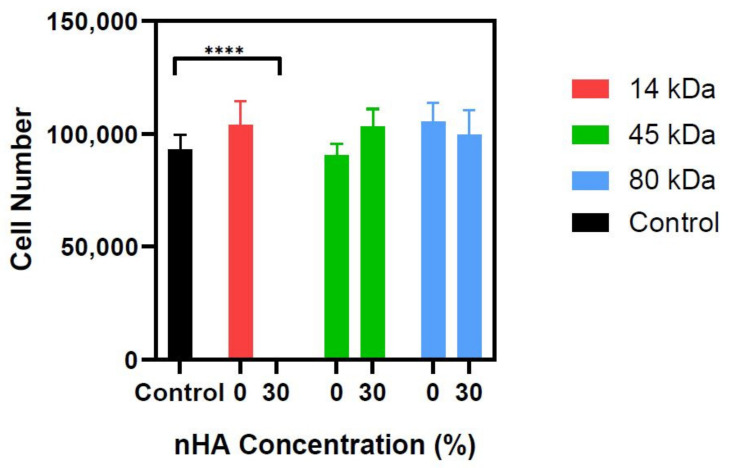
Cytocompatibility study of PCL and PCL-nHA logpiles. The bar graph shows the metabolic activity of MG-63 cells, measured using CellTiter assay, and correlated to the actual cell number via standard curve of cells. Statistical significance as followed: **** is *p* ≤ 0.0001.

## Data Availability

The data presented in this study are contained within the article.

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
