# Peer review of "Characterization of Polycaprolactone Nanohydroxyapatite Composites with Tunable Degradability Suitable for Indirect Printing"

_polymers, 2021, doi:10.3390/polym13020295_

Round 1

Reviewer 1 Report

Comments on the manuscript polymers-1066887, submitted by Doyle et al.

This work deals with characterization of polycaprolactone nanohydroxyapatite composites with tunable degradability. Scaffolds were fabricated by indirect 3D printing. In general, the paper is interesting, it is well-written, but some issues should be resolved before publication.

Comments.

1.- Authors used PCL with different molecular weights to create a material with a tunable degradation rate. However, the time used in the degradation studies (Fig. 5) was only 3 days (72 h). This is a very short time to observe changes in PCL, even in those samples prepared with PCL of MW 14 kDa. The weight loss observed in some formulations are due to disintegration of samples (non-degradation). Please discuss it.

2.- In figure 5, results indicate that samples containing nHA at 30% exhibited the highest “degradation”, except in sample prepared with PCL MW 45 kDa, where sample containing nHA 20% exhibited highest “degradation”. Why? Please explain.

3.- As expected, samples prepared with PCL MW 14 kDa, presented poor fabricability. Why did authors use PCL with this low MW.?

4.- In a clinical use, scaffolds with potential use in bone regeneration, should fulfill with suitable compressive properties. I suggest authors report this parameter and not only compressive modulus.

5.- Thermogravimetric analysis carried out under air atmosphere in order to estimate the nHA content is not adequate. I suggest re-calculate the nHA content under nitrogen atmosphere. Air could provide incorrect data.

Author Response

We thank the reviewer for their valuable comments and address their specific observations as follows:

Reviewer 1

  1. Authors used PCL with different molecular weights to create a material with a tunable degradation rate. However, the time used in the degradation studies (Fig. 5) was only 3 days (72 h). This is a very short time to observe changes in PCL, even in those samples prepared with PCL of MW 14 kDa. The weight loss observed in some formulations are due to disintegration of samples (non-degradation). Please discuss it.

We appreciate the concern with regards to the short time frame of the accelerated degradation experiment. Indeed, natural degradation of PCL in vitro or in vivo is a slow process taking months or even years. Importantly, our degradation experiment was conducted under the accelerated conditions (in an alkaline bath). While the timescales of the degradation are shorter, relative trends in degradation rate should still hold between different PCL compositions. We designed our degradation rate experiment based on the previously published protocols (i.e. Lam et al., Biomed. Mater 2008).

To specifically address this comment, we have now added new discussion (lines 307-316) of the previously published studies of similar materials. We trust this discussion clarifies and justifies our choice of the experimental conditions.

We also appreciate the reviewer’s concern regarding disintegration vs degradation. To examine this question, we now included images of the representative samples from each group at the final time point of degradation (Figure 5d). Most of the samples retained their disc shaped geometry, indicating uniform degradation at the surface and with no evidence of disintegration. The three fast degraded samples (14kDa 20%nHA, 14kDa 30%nHA and 80kDa 30%nHA) did show minor irregularity, and while this may be consistent with a degree of disintegration, by these time points the samples had become extremely powdery to the point they could not be handled without some breaking.

We have added discussion of this point to the manuscript in lines 329-334.

  1. In figure 5, results indicate that samples containing nHA at 30% exhibited the highest “degradation”, except in sample prepared with PCL MW 45 kDa, where sample containing nHA 20% exhibited highest “degradation”. Why? Please explain.

We thank again the reviewer for noticing this unusual result. With this critical comment in mind, we performed new viscosity and TGA measurements aiming to double check and verify the composition of the vials of 45 kDa PCL 20% nHA and 45 kDa 30% nHA groups used in the accelerated degradation study (i.e. in case a sample mix-up had occurred).

With the viscosity test, we utilized the large differences seen in the 45k PCL group (Figure 2). From this we observed that the 45 kDa 30% nHA composite was in the same range as when originally tested: the viscosity of this material (at 70 °C) was recorded at 11,220.5 ± 732.0 Pa.s and when retested it was 9525.0 ± 580.4 Pa.s. In comparison, the 45 kDa PCL 20% nHA material originally had a much lower viscosity of 3093.4 ± 580.1 Pa.s. However, when the 45 kDa PCL 20% nHA material was retested, there were discrepancy in the viscosity results from the results originally obtained.

The results of these new measurements showed the following:

--the ‘45 kDa 30% nHA’ vial was correctly labelled, this material exhibited similar viscosity and nHA content to the previously characterized 45 kDa 30% nHA material.

--the ‘45 kDa PCL 20% nHA’ vial was incorrectly labelled, this material showed a large discrepancy in the viscosity results compared to the original tests, and the TGA results showed a far higher nHA content (34%) compared to the original tests (which showed a 22% nHA content).

Based on our investigations, we cannot be sure as to the contents of this particular ‘45 kDa PCL 20% nHA’ vial. This vial was used for the compression and accelerated degradation experiments only. No other measurements are affected. Considering its unknown origin, we have taken the necessary step to remove the data related to this particular vial from the manuscript.  The section has been revised.

  1. As expected, samples prepared with PCL MW 14 kDa, presented poor fabricability. Why did authors use PCL with this low MW?

We deliberately selected a wide range of molecular weights for PCL to assess trends in relevant properties, such as fabricability. While the higher molecular weight PCLs are more commonly used in the published literature, PCLs as low as 10 kDa have been reported for use in bone tissue engineering applications (K.C. Ang et. al “Compressive properties and degradability of poly(e-caprolatone)/hydroxyapatite composites under accelerated hydrolytic degradation”), therefore we intended to incorporate and evaluate the large range of molecular weight of PCLs.

To specifically address this comment, we have added additional lines to the introduction explaining our rationale (lines 86-91).

  1. In a clinical use, scaffolds with potential use in bone regeneration, should fulfill with suitable compressive properties. I suggest authors report this parameter and not only compressive modulus.

We agree that when creating a functional implant it is not enough to only report on the material’s compressive modulus. For this, a full analysis of mechanical properties such as tensile strength, fracture toughness, sheer modulus, stiffness, fatigues strength or failure mechanism as well as dynamic testing will be required. However, the aim of this initial experiment was to screen the composite materials and obtain comparative results to determine which composite would be the most suitable with the application of bone scaffolds in mind. Detailed analysis of the mechanical properties of particular scaffolds is outside the scope of this study.

To address this comment, we have clarified our rationale by adding a new text (lines 255-258) to the section on mechanical testing.

  1. Thermogravimetric analysis carried out under air atmosphere in order to estimate the nHA content is not adequate. I suggest re-calculate the nHA content under nitrogen atmosphere. Air could provide incorrect data.

We appreciate this comment and address it as follows.

When looking at the literature specifically concerned with PCL/HA composites, studies vary between those that use either nitrogen or an inert gas, air or simply do not report which gas was used in experimentation. Furthermore, we would like to refer to the paper by Liao et. al. “Thermal decomposition and reconstitution of hydroxyapatite in air atmosphere” which specifically looked at the properties of hydroxyapatite (HA) in the TGA using air instead of nitrogen. HA decomposes by first releasing OH- ions making OHA, then the OHA decomposes into tetracalcium phosphate (TTCP) and α-tricalcium phosphate (αTCP). Liao et. al found the gradual release of OH- ions to occur between 1000-1350 °C, while the breakdown of OHA into TTCP and αTCP only occurs above 1360 °C. Additionally, with our control PCL groups we observed a complete burn off of the PCL by completion of the test at 850 °C. Therefore, we can be confident that in our TGA testing in the air environment the PCL is completely removed leaving only nHA, and that the nHA has not begun to breakdown at the completion of the TGA testing at 850 °C.

Reviewer 2 Report

In this paper by Doyle et al., the authors synthesized polycaprolactone nanohydroxyapatite composites and characterized their biodegradability profile. The results are very interesting and definitely fit the scope of Polymers. Most of the findings stand on a solid foundation of results, therefore no considerable objections to this study can be described. However, there is some room for improvement. Please find the suggestions below:
1) It would be good to stress in the "in this work" section if such an approach was not conducted by others if it is the case. A precise description of the novelty factor helps the readers to understand the impact of a paper. It is often decisive whether to read it or not.
2) The experimental section should be improved to reach a sufficient level of scientific rigor. Only then, the study becomes reproducible and it is very important to enable others to build on these findings. For instance, the flow rate of air in TGA was not given and it often influences the shapes of the thermograms a lot. Besides, what was the sample weight?
3) Headlines should not be separated from the corresponding sections (e.g. Line 191).
4) Captions should also accompany figures (Lines 290-292).
5) The arrangement of the text should be improved. For example, there is empty space on Pages 8 and 10.
6) Trends of degradation presented in Fig. 5 are interesting. and the images of the samples in Figs. 6 and 7. are quite informative. I wonder if the authors possess the corresponding pictures of samples after various degradation times shown in Fig. 5. It would be useful to include them.
7) Conclusions section should include a description of the impact and future outlook.

Author Response

We thank the reviewer for their valuable comments and address their specific observations as follows:

Reviewer 2

  1. It would be good to stress in the "in this work" section if such an approach was not conducted by others if it is the case. A precise description of the novelty factor helps the readers to understand the impact of a paper. It is often decisive whether to read it or not.

We thank Reviewer 2 for this comment. We have revised Abstract and Conclusion sections (lines 21-23 and 426-429, respectively) to address the recommendation and highlight the novelty of our work. We believe that this level of characterization of both PCL molecular weights and HA content has not been performed and reported as yet.  

  1. The experimental section should be improved to reach a sufficient level of scientific rigor. Only then, the study becomes reproducible and it is very important to enable others to build on these findings. For instance, the flow rate of air in TGA was not given and it often influences the shapes of the thermograms a lot. Besides, what was the sample weight?

We thank Reviewer 2 for this observation. The methodology section was amended, the TGA method and experimental protocol were revised to include more detailed descriptions as requested.

  1. Headlines should not be separated from the corresponding sections (e.g. Line 191).

The amendments were made as requested.

  1. Captions should also accompany figures (Lines 290-292).

We have fixed these errors in the manuscript.

  1. The arrangement of the text should be improved. For example, there is empty space on Pages 8 and 10.

The amendments were made as requested.

  1. Trends of degradation presented in Fig. 5 are interesting and the images of the samples in Figs. 6 and 7. are quite informative. I wonder if the authors possess the corresponding pictures of samples after various degradation times shown in Fig. 5. It would be useful to include them.

Thank you for this comment. As recommended, we have added Figure 5d to show the degraded samples at the final time point.

  1. Conclusions section should include a description of the impact and future outlook.

As requested, we have added lines 426-429 and 438-443 in the Conclusion section to highlight the impact of our work as well as the possible future outlook.

Round 2

Reviewer 2 Report

Thank you for responding to my concerns. I recommend the publication of the article.

Please ensure at the proof stage that:
- headlines are not separated from corresponding paragraphs (Lines 96 and 97 as well as 426)
- captions accompany the corresponding figures (Lines 234-237)